# Understanding Functional Neurological Disorder: Recent Insights and Diagnostic Challenges

**DOI:** 10.3390/ijms25084470

**Published:** 2024-04-18

**Authors:** Ioannis Mavroudis, Dimitrios Kazis, Fatima Zahra Kamal, Irina-Luciana Gurzu, Alin Ciobica, Manuela Pădurariu, Bogdan Novac, Alin Iordache

**Affiliations:** 1Department of Neuroscience, Leeds Teaching Hospitals, NHS Trust, Leeds LS2 9JT, UK; i.mavroudis@nhs.net; 2Faculty of Medicine, Leeds University, Leeds LS2 9JT, UK; 3Third Department of Neurology, Aristotle University of Thessaloniki, 541 24 Thessaloniki, Greece; dimitrios.kazis@gmail.com; 4Higher Institute of Nursing Professions and Health Technical (ISPITS), Marrakech 40000, Morocco; 5Laboratory of Physical Chemistry of Processes and Materials, Faculty of Sciences and Techniques, Hassan First University, Settat 26000, Morocco; 6Faculty of Medicine, University of Medicine and Pharmacy “Grigore T. Popa”, 700115 Iasi, Romaniaaliniordache@yahoo.com (A.I.); 7Department of Biology, Faculty of Biology, Alexandru Ioan Cuza University of Iasi, Carol I Avenue 20th A, 700505 Iasi, Romania; 8Center of Biomedical Research, Romanian Academy, Iasi Branch, Teodor Codrescu 2, 700481 Iasi, Romania; 9Academy of Romanian Scientists, 3 Ilfov, 050044 Bucharest, Romania; 10Preclinical Department, Apollonia University, Păcurari Street 11, 700511 Iasi, Romania; 11“Socola” Institute of Psychiatry, Șoseaua Bucium 36, 700282 Iasi, Romania; manuelapadurariu@yahoo.it

**Keywords:** functional neurological disorder, neuroimaging, pathophysiology, epidemiology, phenotypic heterogeneity, differential diagnosis

## Abstract

Functional neurological disorder (FND), formerly called conversion disorder, is a condition characterized by neurological symptoms that lack an identifiable organic purpose. These signs, which can consist of motor, sensory, or cognitive disturbances, are not deliberately produced and often vary in severity. Its diagnosis is predicated on clinical evaluation and the exclusion of other medical or psychiatric situations. Its treatment typically involves a multidisciplinary technique addressing each of the neurological symptoms and underlying psychological factors via a mixture of medical management, psychotherapy, and supportive interventions. Recent advances in neuroimaging and a deeper exploration of its epidemiology, pathophysiology, and clinical presentation have shed new light on this disorder. This paper synthesizes the current knowledge on FND, focusing on its epidemiology and underlying mechanisms, neuroimaging insights, and the differentiation of FND from feigning or malingering. This review highlights the phenotypic heterogeneity of FND and the diagnostic challenges it presents. It also discusses the significant role of neuroimaging in unraveling the complex neural underpinnings of FND and its potential in predicting treatment response. This paper underscores the importance of a nuanced understanding of FND in informing clinical practice and guiding future research. With advancements in neuroimaging techniques and growing recognition of the disorder’s multifaceted nature, the paper suggests a promising trajectory toward more effective, personalized treatment strategies and a better overall understanding of the disorder.

## 1. Introduction

Functional neurological disorder (FND) represents a complex and multifaceted condition situated at the intersection of neurology and psychiatry. Historically known as conversion disorder or hysteria, FND has garnered significant interest due to its prevalence and the disabling impact it has on patients. Early clinical neuroscience leaders, including Jean-Martin Charcot, were intrigued by FND, recognizing its significance alongside other neurological and mental conditions [1]. However, due to limitations in neuroscientific tools and a subsequent divide between neurology and psychiatry, FND became a borderland condition, often misunderstood and under-researched throughout much of the 20th century.

The last two decades have witnessed a renaissance in FND research, driven by advances in the diagnostic criteria, treatment modalities, and, notably, neuroimaging techniques. This renewed focus has helped reshape the understanding of FND, moving from solely psychological conceptualizations to a more integrated biopsychosocial framework [2,3,4]. The increasing prevalence of FND, recognized as the second most common outpatient neurological diagnosis, underscores the urgency for a deeper understanding of this disorder [3,5].

Recent studies have revealed varying rates of prevalence and incidence for functional neurological disorder (FND) across different populations, underscoring the significant burden and diversity in its distribution. FND is estimated to comprise at least 5% to 10% of new neurological consultations, ranking as the second most common reason for visiting a neurologist after headache. Conservatively estimated at 12 cases per 100,000 people per year, FND results in approximately 8000 new diagnoses annually in the UK, with an estimated 50,000 to 100,000 individuals affected in the community. However, these figures likely underestimate its true prevalence due to underdiagnosis and misdiagnosis, particularly in regions with limited access to specialized care or diagnostic resources [6].

Yong et al. (2023) conducted a study spanning 36 months at a regional children’s hospital, revealing an annual incidence of 18.3 per 100,000 children for functional neurological disorder (FND) [7]. This finding stands in contrast to the typical onset of FND in early to mid-adulthood, where the peak occurrence usually arises in the third and fourth decades of life [8]. Among the 97 children diagnosed with FND, aged between 5 and 15 years, a noteworthy 70% were female, with a median age of onset at 13 years [7]. This aligns with the findings of a one-stage meta-analysis conducted by Lidstone et al. (2023), indicating a disproportionate impact on women across FND phenotypes, with its rates consistently hovering around 70% in most large-scale studies [9].

Geographically, FND prevalence varies, with higher rates reported in industrialized nations compared to developing countries [10,11]. This disparity may reflect differences in healthcare infrastructure, access to mental health services, cultural attitudes toward neurological and psychiatric conditions, and diagnostic practices. Furthermore, socioeconomic factors, such as low financial security, income, and educational attainment, have been associated with an increased risk of FND, highlighting the complex interplay between social determinants of health and disease susceptibility.

The epidemiology of FND reveals its widespread impact, yet there remains a significant gap in understanding its precise mechanisms. This gap extends to the difficulties in differentiating FND from feigning or malingering, a challenge that has historically contributed to stigma and barriers in diagnosis and treatment [3,4,5]. Neuroimaging studies have started to unveil the complex neural circuitry involved in FND, pointing towards a multi-network brain disorder implicating limbic, salience, self-agency, multimodal integration, and attentional and sensorimotor circuits [3]. These findings have been pivotal in reshaping the narrative around FND, providing a biological basis for a disorder historically marred by misconceptions and oversimplifications.

This review aims to synthesize the current knowledge on the epidemiology, mechanisms, and role of neuroimaging in FND. It also seeks to clarify the distinction between FND and feigning or malingering based on recent scientific evidence and clinical insights. Through an analysis of the contemporary research, including neuroimaging studies, this paper will contribute to a more nuanced understanding of FND, aiming to inform clinical practice and guide future research in this evolving field.

## 2. Etiology of Functional Neurological Symptom Disorder (FND)

The etiology of Functional Neurological Symptom Disorder, formerly known as conversion disorder, is multifaceted, involving various biological, psychological, and social factors. These factors are more prevalent in patients with FND than in those with comparable symptoms due to recognized diseases [10,11,12,13]. However, it is important to note that while these factors may contribute to the disorder, they are not individually causal.

Several neurobiological factors contribute to functional neurological disorder (FND). In patients with FND, abnormalities have been reported in neurotransmitters such as dopamine, serotonin, and gamma-aminobutyric acid (GABA) [14,15]. Moreover, the presence of inflammatory markers and microglial activation in FND patients suggests a possible immune-mediated mechanism of symptom generation [16,17]. FND may also be caused by abnormalities in neuroplasticity, including synaptic plasticity and cortical reorganization, which affect the brain’s ability to adapt to environmental stressors and maintain normal neuronal function [18,19,20].

In addition to neurological factors, psychological factors considerably make a contribution to the pathophysiology of FND. Psychological factors such as stressful life events, interpersonal conflicts, and adverse childhood experiences have traditionally been viewed as potential causes of FND. A meta-analysis of 34 retrospective studies highlighted that stressful life events and maltreatment, including emotional neglect and sexual and physical abuse, are more common in FND patients than in controls [9]. Nonetheless, not all patients with FND report such psychological factors, nor are they specific to FND. Maladaptive cognitive processes, characterized by cognitive distortions and attentional biases, also play a pivotal role in perpetuating FND signs [21,22,23,24]. Moreover, dysregulated emotional processing, like heightened emotional reactivity and alexithymia, has been implicated in FND [22]. Furthermore, environmental stressors like traumatic life events and social adversity affect both the onset and exacerbation of FND signs and symptoms [22,25].

Environmental factors additionally make contributions to FND pathophysiology. Cultural factors and societal attitudes in relation to sickness affect FND’s presentation and management [26,27]. In addition, social support networks and stigma experiences affect the direction of FND [25,28]. Access to healthcare services and the availability of specialized FND treatment programs also have an impact on diagnosis and management [29].

FND is often associated with pre-existing psychiatric disorders (like depression, anxiety, and personality disorders), other somatic conditions (such as pain and fatigue), and functional somatic disorders like irritable bowel syndrome. Neurological illnesses and physical injuries are also common precursors. Additionally, FND has been linked with lower socioeconomic status [30,31,32,33,34,35,36,37].

## 3. The Role of Personality Traits in Functional Neurological Disorder (FND)

Functional neurological disorder (FND) has historically been linked to various etiological factors, including psychological stressors and adverse childhood events. However, empirical testing of these associations has been relatively limited. The DSM-5 no longer requires a recent stressor as a diagnostic criterion for FND, acknowledging that functional motor symptoms may occur with or without identifiable stressors [30,38,39].

In a controlled study of functional motor disorders, including functional limb weakness, a higher frequency of adverse childhood and adult experiences was noted in patients compared to control subjects. However, it is crucial to note that more than half of the participants reported no experience of abuse, with sexual abuse being relatively infrequent. Adverse experiences were found in three to four case patients compared to two in ten control subjects, whether assessed using an interview or a questionnaire [40,41,42,43].

A systematic review and meta-analysis of 34 controlled studies of stressful life events and childhood maltreatment in FND highlighted emotional neglect as a clear risk factor, but it is also important to recognize that these experiences are not present in the majority of patients and occur at similar frequencies in individuals with psychiatric disorders and other conditions, including migraines [38,39]. Additionally, the role of birth order as a predisposing factor for FND has been mooted but not supported by data. Studies have shown no differences in birth order among patients with FND, suggesting that birth order is not a significant predisposing factor [44,45].

Personality traits have long been associated with functional motor disorders, often linked to the concept of a hysterical or histrionic personality. Studies focusing on personality disorders, especially cluster B disorders, found them to be more common in FND but still only present as a minority. The reported frequencies vary widely [32,46,47]. Stone et al., in a prospective case–control study, found small to medium effect sizes for higher neuroticism and lower openness in FND, correlating with higher frequencies of depression and anxiety in these patients [33]. Previous studies on personality traits in FND have shown mixed results, indicating no consistently found personality traits in these patients compared to control subjects [48,49].

Medical and surgical comorbidities present a clearer pattern in FND. Higher rates of surgical procedures and a history of neurological and other disease diagnoses have been recognized as important risk factors. The high rate of sterilization, particularly compared to gender-matched controls, might suggest a greater willingness to undergo surgical procedures in FND patients [34,50,51,52,53,54,55,56,57]. Despite these patterns, the concept of symptom modeling in the environment as an etiological factor for FND, going back to Janet’s idée fixe, remains a challenging and arguably unfalsifiable hypothesis. Studies exploring disease modeling have not found it to be a diagnostic feature of FND, nor have they conclusively demonstrated its etiological significance [58,59].

Stressful events at work can generate a psychological tension that, associated with traumatic experiences, contributes to the appearance or worsening of FND symptoms in susceptible employees [18]. Vanini G et al. [60] illustrated that this cluster of symptoms is frequent in health workers who work in a high-stress, conflictual environment with multiple tasks and tight deadlines. Reciprocal understanding and support in the workplace from both employers and employees are essential to effectively manage these challenges and find solutions that allow workers with FND to manage their symptoms and be fulfilled in their work [60].

While FND has been historically linked to various predisposing factors, including adverse childhood experiences and certain personality traits, the overall picture emerging from recent studies is that these factors, while significant, do not present in the majority of patients. The etiology of FND appears to be multifactorial, with no single predisposing factor universally present in all cases.

## 4. Pathophysiological Mechanisms and Suggested Pathogenic Models of Functional Neurological Disorder

The pathophysiology of functional neurological disorder (FND) is multifaceted, involving complex interactions across various neural networks. Recent advancements in neuroimaging have been pivotal in elucidating these mechanisms, revealing alterations in limbic, salience, self-agency, multimodal integration, attentional, and sensorimotor circuits (Table 1).

Altered Limbic and Salience Network Activity: A hallmark of FND is altered activity within the limbic and salience networks (Table 1). Studies using functional MRI (fMRI) have consistently shown increased limbic/paralimbic activity in patients with FND compared to controls [61]. Specifically, impairments in amygdala habituation and increased sensitization are noted in patients with functional motor symptoms, along with heightened functional connectivity between the amygdala and motor control circuits [62,63]. These findings suggest an augmented limbic influence over motor behavior, which might contribute to the motor manifestations of FND. However, it is important to note that the findings have been inconsistent, with some studies reporting normal or hypoactive amygdala responses in certain FND subtypes, such as functional movement disorder (FMD) and functional [psychogenic nonepileptic/dissociative] seizure (PNES) cohorts [64,65,66,67,68]. 

Self-Agency and Multimodal Integration Disruptions: Another aspect of FND’s pathophysiology involves disruptions in the networks related to self-agency and multimodal integration (Table 1) [63]. Task-based neuroimaging studies have identified abnormalities in brain activations during conditions under which the perception of voluntary control over movements is altered in FND patients [68,69]. This disruption may contribute to the experience of involuntary movements, a common feature of FMD.

Attentional and Sensorimotor Circuit Alterations: The sensorimotor and attentional networks also play a critical role in FND (Table 1) [70]. Altered insula and cingulate gyrus activations have been documented across emotion processing and motor control tasks [71]. These areas are crucial for integrating sensory information and emotional processing, suggesting that their dysregulation might underlie some of the clinical manifestations of FND.

FND englobes signs that affect motor, sensory, and cognitive functions and regularly results in significant impairment of daily activities and overall well-being [85]. FND’s hallmark characteristic is the presence of neurological signs and symptoms that cannot be defined by way of underlying organic pathology, leading to a diagnosis based on exclusion criteria and positive clinical signs [86].

In FND, its motor signs and symptoms can appear in diverse forms, such as weakness tremors, abnormal movements, gait disturbances, and paralysis [87]. Weakness is one of the most familial motor symptoms visible in FND and generally manifests as weakness affecting one or more limbs or even the complete body. This weakness is typically inconsistent and variable, fluctuating in severity and distribution over time. In FND, tremors may also resemble those seen in movement disorders, including Parkinson’s disorder or essential tremor; however, they lack the characteristic patterns and reactions to medication [87]. Abnormal movements, like jerking or shaking, may additionally arise and can mimic epileptic seizures or other hyperkinetic motion disorders [88]. Gait disturbances may additionally manifest as unsteady or uncoordinated walking patterns, often leading to falls or trouble with maintaining balance [89]. In extreme cases, patients can also experience functional paralysis, wherein they are not able to move certain body parts notwithstanding intact motor function [90]. 

The sensory symptoms encompass a variety of abnormalities in FND, inclusive of altered sensation, numbness, tingling, and sensory loss. Patients may also record uncommon sensations including pins and needles, burning, or electric shocks in diverse parts of the body [91]. These sensations regularly lack a clear dermatomal or peripheral nerve distribution and may be inconsistent with or disproportionate to any identifiable peripheral pathology [91]. Numbness and tingling sensations can also have an effect on particular regions or unfold diffusely, on occasion alternating among unique body regions [92]. Sensory loss can include any modality, inclusive of contact, temperature, or proprioception, and can be temporary or chronic [93]. Its cognitive symptoms are also less identified; however, they can significantly impact daily functioning and quality of life. Patients may experience cognitive impairments such as attention deficits, memory difficulties, executive dysfunction, and language disturbances [94]. Memory problems may manifest as gaps in recall or difficulty in retaining new information, often leading to frustration and anxiety [95]. Attention deficits can also result in distractibility, difficulty concentrating, or problems with sustained focus on tasks [96]. Executive dysfunction can affect planning, organization, and problem-solving abilities, impairing an individual’s ability to initiate and complete tasks effectively [97]. Language disturbances may include difficulties in word-finding, speech production, or understanding language, resembling the aphasic symptoms seen in neurological conditions [98]. 

The numerous array of symptoms seen in FND can have profound repercussions on daily activities, occupational functioning, and normal well-being. Its motor signs may additionally restrict mobility and independence, affecting activities of daily living like dressing, grooming, and driving. Its sensory signs and symptoms may also disrupt sensory processing and integration, leading to problems in decoding and responding to environmental stimuli [99,100,101]. Its cognitive symptoms can impair cognitive functioning and decision-making abilities, impacting work performance, social interactions, and interpersonal relationships [102,103]. The cumulative effect of these symptoms can also make a contribution to tremendous distress, incapacity, and a reduced satisfaction with life for FND individuals and their caregivers. 

Functional neurological disorder (FND) englobes a huge spectrum of clinical manifestations, ranging from motor, sensory, and cognitive signs and symptoms to disturbances in recognition and autonomic function [86,104]. FND’s phenotypic displays can range extensively amongst individuals, with differences located in the symptom severity, progression, and response to treatment [9,105]. Understanding this diversity is vital for tailoring interventions to cope with the unique needs and challenges of each patient. 

Symptom severity variability: FND’s symptom severity can vary from mild to severe, with few individuals experiencing intermittent or mild signs that do not significantly impair daily functioning, while others may have profound and debilitating symptoms that critically effect their quality of life [106]. Motor signs and symptoms like weakness, tremors, or paralysis may fluctuate in intensity over time, with the durations of remission or exacerbation influenced by factors like stress, emotional state, or environmental triggers [6,107,108]. Sensory symptoms like numbness, tingling, or sensory loss may additionally vary in severity and distribution, affecting different body regions or modalities [109,110]. 

Symptom progression: FND’s symptom progression is highly variable and may follow unpredictable patterns over time. Some individuals may experience a gradual improvement in or the resolution of signs and symptoms with time, while others may have also a chronic or relapsing-remitting course characterized by recurrent episodes of symptom exacerbation [111]. FND’s symptom progression can be stimulated by factors like psychological distress, traumatic experiences, or modifications in psychosocial circumstances, highlighting the complex interaction among biological, psychological, and social factors in the disorder’s trajectory [18,22,112].

Treatment response: The treatment response in FND can also vary broadly among individuals, with some patients displaying a significant improvement with focused interventions, while others might have a limited or partial reaction. The treatment procedures for FND typically contain a multidisciplinary approach tailored to coping with the specific needs and challenges of each patient. Cognitive behavioral therapy (CBT) [113], physiotherapy, occupational therapy, speech therapy, and pharmacotherapy can be applied alone or in combination with target symptom management, functional rehabilitation, and psychosocial support [114,115,116]. 

Personalized intervention approach implications: FND’s numerous phenotypic presentations underscore the importance of personalized intervention approaches that take into account the individual variability in symptom severity, progression, and treatment response. Personalized interventions might also require a complete evaluation of each patient’s unique clinical profile, consisting of physical, psychological, and social factors contributing to their symptoms [106]. This assessment can assist in identifying specific treatment goals and tailoring interventions to addressing the underlying mechanisms driving symptom expression.

## 5. Suggested Pathogenic Models of FND

Cognitive-behavioral models suggest that its functional neurologic symptoms may result from cognitive, emotional, and behavioral influences at the subconscious levels of processing [72,73,74,75]. A cognitive-behavioral perspective posits that the genesis of functional neurologic symptoms is often rooted in the subconscious processing of perception and behavior. One model within this framework suggests that these symptoms can emerge from cognitive, emotional, and behavioral factors operating at subconscious levels [72]. Heightened anxiety and vigilance might activate this mental representation to a degree where it supersedes actual sensory input, consequently distorting perception and behavior [76].

Another cognitive behavioral theory revolves around dissociation in functional neurological symptom disorder [73]. Dissociation is experienced subjectively as a detachment from oneself (depersonalization) or the environment (derealization). This state involves altered awareness and integration of thoughts, feelings, memories, and identity, as well as a disruption in the integration of bodily experiences and functions. During dissociation, patients might experience a loss of motor control or sensory awareness [117]. This dissociative state can be triggered by various factors, such as fatigue, panic attacks [118], physical injuries [119], recognized diseases or pain [120,121], general anesthesia [118,121], or drug side effects [37]. Within this model, symptoms like paralysis or abnormal movement emerge during dissociation, accompanied by a loss of personal connection to bodily movements. The focus on these symptoms, coupled with the fear of their potential implications, might lead to more localized depersonalization, thereby extending the duration of the symptoms. In the case of functional seizures, the initial symptoms of autonomic arousal may escalate to the point where the patient’s response is a loss of awareness, appearing as a blackout [75,122,123,124]. Research on patients with functional neurological symptom disorder indicates they might have a reduced conscious awareness of their emotional symptoms, like anxiety, which may explain their tendency to report the physical rather than emotional aspects of these events [125,126]. 

Neurobiologic models suggest that FND is posited to involve intricate abnormalities in the neural networks, rather than being linked to a singular brain structure anomaly [63]. These networks encompass the orbitofrontal and anterior cingulate cortex and subcortical limbic structures, which may be triggered by stress and other factors. These activated areas are hypothesized to provide inputs to the inhibitory basal ganglia–thalamocortical circuits, leading to a reduction in conscious motor or sensory processing [77,78]. Functional MRI studies comparing FND patients and healthy controls have revealed differences in regional brain activity during recall of traumatic events, such as altered activation in the prefrontal cortex and hippocampus, with some studies showing a correlation between these brain activities and symptom severity [79,80]. Aberrations in the brain networks, including those encompassing the inferior parietal lobe and the temporoparietal junction, which are crucial to self-agency perception, have been observed [127,128]. Hypersensitive amygdala responses to fear stimuli, coupled with abnormal self-focused attention, such as depersonalization during injury or panic, may produce sensations or movements that are perceived as involuntary symptoms due to an altered sense of self-control [73,76].

Structural changes in the brain have also been suggested in neuroimaging studies of FND patients. Structural MRI research comparing patients with controls has found evidence of altered brain structures, such as an increased thalamic volume and reduced sensorimotor cortical thickness [129,130]. However, it remains unclear whether these structural differences are causal factors, confounding comorbidities, or consequences of the disorder.

Psychodynamic models, originating from the classic psychodynamic hypothesis of conversion disorder, suggest that unconscious conflicts manifest as somatic symptoms, serving as a defense against anxiety and distress by keeping the conflict subconscious [81,82,83]. The symptom symbolically represents the conflict, helping the patient avoid overwhelming situations [84]. Subsequent psychodynamic theories focus on abnormal interpersonal relationships stemming from problematic early life experiences or trauma. Here, physical symptoms, viewed as a coping response to emotional dysregulation, may reenact previous patterns of abnormal behavior in response to new conflicts or traumatic events. 

The Bayesian model offers a theoretical framework that is increasingly used to conceptualize the pathophysiology of functional neurological disorder (FND) [76]. This model is grounded in the principles of predictive coding and active inference, providing insights into how the brain processes information and how this might go awry in FND. According to the Bayesian model, the brain constantly generates predictions about sensory inputs based on prior experiences. These predictions are then updated by incoming sensory information according to a process known as predictive coding. Active inference refers to the brain’s mechanism of minimizing prediction errors by either adjusting its predictions (perceptual inference) or by acting on the environment (active inference) [76].

In FND, there appears to be a disruption in this predictive processing [131]. The model suggests that symptoms arise due to an imbalance in the integration of sensory data and top-down predictions. This disruption can manifest as alterations in perception, sense of agency, and motor control, which are characteristic of FND. The symptoms of FND therefore can be viewed as a result of the brain’s inaccurate predictions about bodily states and motor actions.

The Bayesian model also implicates the processing of emotional and sensory signals in FND. In cases where there is a heightened sensitivity to emotional or interoceptive signals, the brain might generate predictions that are overly influenced by these signals, leading to the characteristic symptoms of FND. This aligns with neuroimaging findings that show altered activity in the limbic and salience networks in FND patients [132]. The Bayesian model is supported by neuroimaging studies that show altered brain activations in areas involved in predictive processing and emotional regulation [132,133]. This integration of theoretical models with empirical neuroimaging data offers a comprehensive approach to understanding and treating FND.

Understanding FND through the lens of the Bayesian model has significant implications for treatment. It suggests that therapeutic interventions might be aimed at recalibrating the brain’s predictive models. For instance, therapies like cognitive behavioral therapy (CBT) may help in modifying maladaptive beliefs and expectations, thereby adjusting the brain’s predictions and reducing symptom severity.

## 6. Diagnosis, Criteria, and Techniques for Functional Neurological Disorder

In the Diagnostic and Statistical Manual of Mental Disorders, Fifth Edition (DSM-5), several criteria need to be met, including symptoms of altered motor or sensory function, inconsistency with recognized neurological or medical conditions, the absence of an alternative explanation, and significant distress or functional impairment in social or occupational domains [134]. The diagnostic process hinges critically on the patient’s history and direct observation. Key areas of inquiry include a thorough examination of all their current symptoms encompassing neurological, psychological, and constitutional aspects. Clinicians should explore the onset circumstances, the illness course, including variability and triggers, the patient’s disability level, their psychosocial functioning, and any dissociative experiences. A comprehensive review of the patient’s family history, past functional symptoms and disorders, previous clinical encounters, recent psychological stressors, and symptoms of comorbid psychiatric disorders is also essential [118,134].

Identifying positive signs is crucial for diagnosing FND. Positive signs are particularly valuable, aiding clinicians in confirming a diagnosis. These signs include internal inconsistency and incongruence with the known abnormal movement patterns observed in other neurological diseases. Internal inconsistency, characterized by symptom variability over time or context, is a common feature across most FND phenotypes. Specificity for positive signs in functional motor and sensory symptom disorders ranges widely in sensitivity but generally remains high, with most signs having over 90% specificity [118].

The use of distracting maneuvers can temporarily diminish or suppress FND’s symptoms. Cognitive distractions might involve complex mental tasks, while motor distractions include activities that engage body parts not affected by the abnormal movement. However, it is noteworthy that these techniques might be less effective in longstanding FND cases or certain subtypes like fixed dystonia, functional myoclonus, or episodic FND.

A defining characteristic of FND is the variability of its symptoms, which may fluctuate in frequency, amplitude, direction, or location. This variability might be observed over the natural course of the condition or during a single assessment, with its symptoms often changing with body position and environment.

Psychiatrists, neurologists, psychologists, neuroscientists, and other allied health expert collaboration plays a pivotal function in FND research and improving clinical care [6]. Neurologists, by means of their knowledge of the brain structure and function, diagnose and manage the neurological aspects of FND and assist in identifying the signs of FND and distinguishing them from different neurological situations [135]. Psychiatrists carry expertise in dealing with comorbid psychiatric signs and symptoms, making use of psychotherapeutic interventions, and handling the psychosocial elements contributing to FND [7]. Psychologists make contributions in terms of the behavioral and cognitive aspects of FND. They provide psychological evaluations and interventions (e.g., cognitive behavioral therapy) and address factors like trauma, stress, and coping techniques [7]. Neuroscientists inspect the neural mechanisms underlying FND [136]. Via neuroimaging research, electrophysiological exams, and animal models, neuroscientists elucidate the brain circuits and neurochemical procedures involved in FND’s pathophysiology [137]. Allied health professionals, consisting of physical therapists, occupational therapists, speech–language pathologists, and other allied health professionals [138], provide specialized interventions to deal with motor, sensory, and cognitive signs in and optimize the functional abilities and improve the quality of life of FND individuals [139,140]. This collaboration enhances the treatment effectiveness, diagnostic accuracy, and patient results. Patients advantage from comprehensive and coordinated care tailored to their needs (Figure 1).

Here is a model of guidance for healthcare professionals engaged in the assessment and management of patients with functional neurological disorder (FND), available at https://fndaustralia.com.au/resources/FND-Learning-guide-for-nurses.pdf (accessed on 10 April 2024). To optimize the medical outcomes, healthcare specialists need to embrace a multi-faceted method that consists of powerful affected patient education, organizing sturdy therapeutic alliances, and coordinating care across disciplines. Patient education should entail presenting personalized records in comprehensible language, complemented by visual aids and bolstered learning opportunities to allow patients to play an active role in their care. Developing a therapeutic consensus depends on active listening, collaborative decision-making, and establishing a trusting relationship, while acknowledging the significance of the patient’s support system. Coordinating multidisciplinary care requires the formation of a cohesive team with clear communication channels, defined roles, and patient-centered care plans, ensuring comprehensive and holistic support. By implementing these strategies, healthcare professionals can empower patients, strengthen therapeutic relationships, and improve the quality and efficiency of care delivery, ultimately translating into improved clinical outcomes.

## 7. Distinguishing Functional Neurological Disorder from Deliberate Symptom Fabrication (Feigning or Malingering)

Functional neurological disorder (FND) is often misinterpreted as Deliberate Symptom Fabrication (feigning or malingering). However, recent research and clinical observations strongly differentiate FND from these volitional conditions (Table 2).

For instance, a study utilizing fMRI has shown differences in the brain activation patterns between patients with genuine FND and those feigning weakness. Patients with genuine weakness exhibit reduced activation of the cortical areas related to hand movement compared to controls when their movement is observed, suggesting impairment of movement conceptualization rather than active inhibition from the frontal lobe. In contrast, feigning controls demonstrate similar patterns of activity in both no-go and go trials with the feigned weak hand, indicating intentional mimicry rather than genuine weakness [141].

FND has a historical and cross-cultural consistency in its presentations, from early medical documents to contemporary reports. This consistency suggests an underlying authentic medical condition rather than a fabricated one. Additionally, the subjective experiences reported by individuals with FND, such as specific symptoms related to functional seizures, are consistent and distinct, further supporting its legitimacy as a genuine disorder [142,143,144]. In contrast to the behavior expected from feigning or malingering, individuals with FND often seek and undergo numerous medical investigations, typically resulting in normal findings unless a comorbid condition exists [4,145,146]. This behavior indicates a genuine pursuit of medical understanding and treatment, inconsistent with feigning or malingering motivations.

The long-term persistence of symptoms in FND and their improvement with specific treatments, such as cognitive behavioral therapy, contradict the pattern expected from feigned conditions [4,147,148]. Improvement is often observed with treatments designed specifically for FND, distinguishing it from feigned symptoms.

Neuroimaging and experimental studies provide evidence of distinctive neural function in people with FND compared to individuals feigning similar symptoms [4,141]. These studies show differences in brain activation and physiological responses, underpinning FND as a neurologically based disorder rather than a product of conscious simulation.

Theoretical models, such as the Bayesian brain model, propose that FND arises from a dysfunction in the brain’s predictive processing, leading to symptoms experienced as involuntary [4,76]. This model, supported by neuroimaging findings, offers a neuroscientific explanation for FND fundamentally different from the conscious production of symptoms seen in feigning or malingering.

A combination of clinical, epidemiological, and experimental data firmly establishes FND as a legitimate medical condition, distinct from feigning or malingering. This distinction is critical for proper diagnosis, reducing stigma, and guiding effective treatment strategies.

## 8. Neuroimaging in Functional Neurological Disorder (FND)

Neuroimaging studies have significantly advanced our understanding of FND, revealing it as a multi-network brain disorder. Various imaging modalities, including functional and structural neuroimaging, have been instrumental in identifying alterations in the limbic/salience, self-agency/multimodal integration, and attentional and sensorimotor circuits (Table 3) [63,68,70,132]. 

Functional MRI (fMRI) studies using affectively valenced tasks have generally shown increased limbic/paralimbic activity in FND patients compared to controls, including altered amygdala habituation and sensitization and increased connectivity between the amygdala and motor control circuits (Table 3) [2,64,71,78,79]. These findings suggest a heightened limbic influence over motor behavior. However, inconsistencies exist, with some studies reporting normal or reduced amygdala activity in certain FND subtypes [66,67]. 

Task-based neuroimaging in FND has also highlighted the role of the right temporoparietal junction (rTPJ) in self-agency disturbances, particularly in functional movement disorders (Table 3) [63,69,127]. 

Resting-state functional connectivity and quantitative structural imaging have provided insights into the intrinsic brain architecture of FND (Table 3). Increased connectivity between the emotion processing and motor control networks has been observed in resting-state functional connectivity MRI studies, with the symptom severity correlating with increased cingulo-insular coupling to the motor control areas [80,149,150,151]. 

Furthermore, the preliminary data suggest that baseline increased task-related corticolimbic activity may predict treatment responses to cognitive behavioral therapy (CBT) and short-term inpatient multidisciplinary motor retraining [2,152]. 

Gray matter alterations in the sensorimotor, cingulo-insular, and amygdala areas have been identified, although these findings have been inconsistent (Table 3) [2,71,77,153,154]. Subgroup-specific effects, such as a reduced left anterior insula volume in patients reporting severe physical health impairments, have also been demonstrated [19]. 

White matter characterization in FND, while still in its early stages, has indicated alterations in the limbic and associative fiber bundles compared to healthy controls (Table 3) [80,155,156,157]. 

It is crucial to note that the interpretation of neuroimaging findings in FND must consider the phenotypic heterogeneity of the disorder. This includes varying symptom severity, episodic vs. persistent symptoms, the duration and onset of illness, symptom type and overlap, and symptom location. Furthermore, additional physical and mental health diagnoses are common in FND, complicating the interpretation of imaging results [154]. 

The neuroimaging field in FND is evolving, with emerging approaches like machine learning offering potential diagnostic utility. However, challenges remain, including the need to account for comorbidities, medication effects, and the heterogeneity of FND presentations. Future research may benefit from focusing on individual differences, biomarkers of symptom severity, illness duration, and the risk factors for developing FND.

## 9. Discussion

Functional neurological disorder (FND) provides a significant diagnostic challenge because of its phenotypic heterogeneity [154]. FND’s clinical manifestations regularly mimic those of organic neurological disorders, making it tough to distinguish between functional and structural etiologies based entirely on clinical exam or neuroimaging findings. Additionally, FND’s symptoms regularly occur alongside psychiatric comorbidities, which adds complexity to the diagnostic procedure. This necessitates a thorough assessment by multidisciplinary groups to ensure comprehensive care. One of the primary FND diagnostic difficulties is its intersection with a large range of neurological and psychiatric conditions. Motor symptoms like tremors, weakness, and abnormal movements may resemble those seen in neurological issues like multiple sclerosis, Parkinson’s disorder, or epilepsy. Sensory signs and symptoms including numbness, tingling, and sensory loss may additionally mimic peripheral neuropathies or spinal cord lesions. Moreover, cognitive signs which include memory difficulties, attention deficits, and language disturbances may also intersect with the cognitive impairments visible in neurodegenerative diseases or psychiatric problems, which include depression and anxiety. This overlap highlights the importance of a thorough differential diagnosis and careful consideration of its clinical features and natural history and the reaction to treatment in distinguishing FND from other conditions. 

This variability, encompassing symptom severity, episodic versus persistent manifestations, onset, and symptom overlap, complicates the interpretation of neuroimaging findings. Such diversity requires a meticulous and individualized approach in research and clinical settings, emphasizing the need for a tailored diagnostic process that accounts for the wide spectrum of symptom presentations and concurrent health issues. This heterogeneity also underscores the importance of developing more precise and sensitive diagnostic criteria and tools for FND, necessitates individualized diagnostic strategies, and highlights the need for refined criteria that can accommodate the wide spectrum of FND’s presentations. Understanding the heterogeneity in FND is crucial to interpreting neuroimaging findings and tailoring patient-specific treatments.

Neuroimaging has emerged as a key tool in elucidating the pathophysiological underpinnings of FND, revealing alterations in multiple brain networks. Importantly, the preliminary data indicate that neuroimaging might have predictive value in determining the treatment response, particularly in relation to cognitive behavioral therapy (CBT) and multidisciplinary approaches [2,63,68,70,130,152]. This promising avenue suggests that neuroimaging could play a crucial role in personalizing treatment strategies, potentially enhancing the efficacy of therapeutic interventions.

Personalized treatment strategy implementation based on neuroimaging findings faces several obstacles. Neuroimaging data’s complexity demands specialized interpretation skills, often unavailable in standard healthcare settings. Additionally, imaging finding variability within the same diagnosis complicates identifying consistent biomarkers for treatment prediction. The reproducibility and validation of those biomarkers throughout specific populations and imaging modalities continue to be missing, elevating worries about the reliability of making treatment decisions primarily based on neuroimaging. Moreover, the accessibility and value of advanced neuroimaging strategies pose considerable obstacles to widespread adoption, especially in resource-limited environments. Other influencing factors are ethical issues, like patient autonomy and privacy, which must also be carefully navigated to ensure equitable and patient-centered care. Furthermore, longitudinal monitoring is necessary to modify the treatment strategies over time, adding logistical challenges and resource burdens. The streamlined integration of neuroimaging into clinical practice calls for interdisciplinary collaboration and standardized protocols. While neuroimaging studies have identified alterations in brain networks associated with FND, translating these findings into clinical practice remains a complex endeavor. 

The intricate pathophysiology of FND, involving diverse neural networks, highlights the disorder’s complexity. Despite advancements in neuroimaging, fully deciphering the neural mechanisms underlying FND remains a challenge [63,70,130,152]. Future research efforts, particularly those employing large-scale studies and advanced imaging techniques, are essential for a deeper understanding of FND. Collaborative research endeavors could provide critical insights into the disorder’s multifaceted nature and aid in the development of targeted, effective treatments.

Recognizing the perspectives and views of people living with functional neurological disorder (FND) and their caregivers is essential. By acknowledging and incorporating their views, we can make sure that interventions and techniques are tailored to satisfy their precise wishes, in the end improving the quality of care and support supplied to FND people and their caregivers. Staton et al. [158], through their qualitative study, highlighted patient narratives’ importance in informing research and promoting patient-centered care approaches. 

Internalized stigma and self-doubt: Danielle’s experience illustrates the profound effect of internalized stigma and self-doubt on FND individuals. She expressed the conflict of doubting herself and wondering about the validity of her signs, highlighting the detrimental results of professionals attributing physical symptoms to psychological reasons. This narrative emphasizes the importance of patients’ experiences and addressing their worries about self-blame and shame. 

Selective disclosure to professionals: Isla’s tale sheds light on the hesitancy FND individuals may feel in disclosing records to healthcare professionals because of fear of negative judgment. Isla’s reluctance to share certain signs reflects the pervasive impact of internalized stigma and past negative experiences with healthcare providers. This narrative emphasizes the need to develop a safe and non-judgmental environment in which patients feel comfortable sharing their experiences openly. 

Psychological explanations of perceptions: Katherine’s account highlights the challenges individuals face in accepting psychological explanations of their symptoms, especially when they perceive these explanations as blaming. Her resistance to the concept of psychological triggers emphasizes the importance of clinicians approaching psychological explanations with sensitivity and empathy. This narrative emphasizes the need for a collaborative dialogue between patients and professionals to promote understanding and address misconceptions about psychological interventions.

Having to educate professionals: Elizabeth’s experience reflects the frustration of individuals who find themselves in the position of educating healthcare professionals about FND. Her discomfort with being expected to explain her condition underscores the importance of healthcare providers receiving comprehensive training in FND to avoid further exacerbating patients’ feelings of otherness and mistrust. This narrative underscores the necessity of improving professional knowledge and awareness to enhance the quality of care provided to individuals with FND.

Attunement and trust within the therapeutic relationship: Clara’s narrative exemplifies the transformative effect of an effective therapeutic relationship characterized by attunement and trust. Her account highlights the profound distinction a supportive and empathetic healthcare expert can make in supplying validation and knowledge to FND individuals. This narrative underscores the significance of clinicians cultivating attunement and trust with their patients to facilitate collaboration and promote holistic care. 

## 10. Conclusions

Functional neurological disorder (FND) is a multifaceted condition characterized by a complex interplay of neurological, psychological, and sociocultural factors. Despite its prevalence and impact, it remains a challenge in terms of both diagnosis and treatment due to its heterogeneous presentation and the historical stigma associated with it. Recent advances in neuroimaging have started to unravel the intricate neural mechanisms at play, offering new perspectives on this condition.

These neuroimaging findings, coupled with a growing understanding of FND’s epidemiology and clinical manifestations, are reshaping the approach to its diagnosis and treatment. The potential of neuroimaging to predict the treatment response is particularly promising, suggesting a future where personalized treatment strategies could significantly improve patient outcomes. However, the journey to fully understanding and effectively managing FND is far from complete. Ongoing research, especially in neuroimaging and the integration of multi-disciplinary treatment approaches, is crucial. As our understanding deepens, it is imperative that this knowledge translates into clinical practice, reducing stigma and improving care for individuals with FND.

Future research on functional neurological disorder (FND) holds promise via various avenues [159]. Longitudinal studies are important to understanding FND’s natural history, including the trajectory of symptoms, predictors of prognosis, and factors influencing the treatment outcomes [2,160,161,162]. Researchers can identify patterns of spontaneous remission, symptom fluctuation, or progression of the disorder by following FND individuals over time [163,164,165]. In addition, longitudinal studies can also elucidate the comorbidities, impact of psychosocial factors, and life events on the course of FND [166]. Investigating neuroimaging biomarkers longitudinally can also offer insights into changes in brain structure and function associated with symptom evolution [167]. Collaborative efforts related to multiple research centers are crucial for recruiting large cohorts and ensuring long-term follow-up. While translational research aims to bridge the gap between basic science discoveries and clinical applications, it also advances our knowledge of FND’s pathophysiology and informs the development of novel diagnostic and therapeutic strategies [168]. Preclinical studies using animal models of neurological disorders can elucidate the underlying neurobiological mechanisms and uncover the achievability of therapeutic goals [169]. Translational research can also leverage progressive neuroimaging techniques to validate the findings from preclinical models and translate them into clinical biomarkers for FND diagnosis and prognosis [170]. Collaborations among basic scientists, neuroimagers, and clinicians are critical for translating the laboratory findings into significant interventions for FND patients. Innovative intervention trials focused on precise mechanistic pathways retain promise for improving FND outcomes. These trials can investigate novel pharmacological agents, psychological interventions, or neuromodulatory strategies aimed toward modulating aberrant neural circuits implicated in FND’s pathophysiology. For instance, focus on dysfunctional sensorimotor networks or limbic salience via non-invasive brain stimulation techniques like transcranial magnetic stimulation (TMS) or transcranial direct current stimulation (tDCS) may provide a therapeutic advantage for FND individuals [171,172]. Moreover, personalized treatment approaches based on neuroimaging biomarkers can be evaluated in clinical trials to determine their efficacy and feasibility. Multicenter randomized managed trials (RCTs) with rigorous methodological standards are important to evaluate the safety and effectiveness of novel interventions and establish proof-based treatment guidelines for FND [173].

In conclusion, FND stands at the exciting crossroads of neurology and psychiatry. The continued exploration of its complex nature will not only benefit those affected by FND but also contribute to a broader understanding of the mind–brain interface in health and disease. Encouraging critical appraisal of the present literature on functional neurological disorder (FND) is crucial to advance our knowledge of this complicated condition. Researchers should scrutinize the methodological obstacles, capability biases, and areas wanting similar investigation. Rigorous scientific inquiry is key to ensuring valid findings in FND research. By promoting critical appraisal, we can enhance the quality of the research and contribute to improved diagnostic and treatment strategies for FND.

## Figures and Tables

**Figure 1 ijms-25-04470-f001:**
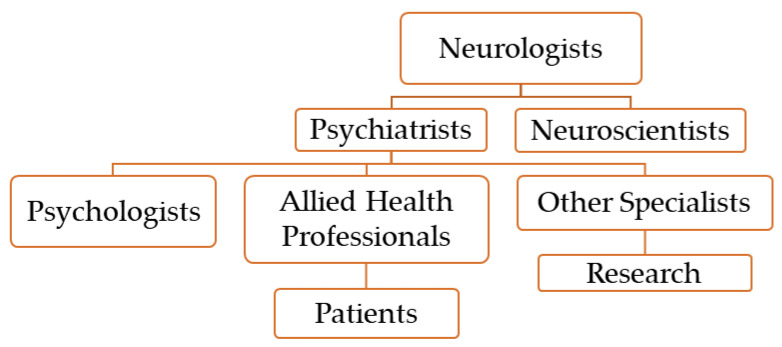
Interdisciplinary collaboration in research and clinical care in FND.

**Table 1 ijms-25-04470-t001:** Summary of pathophysiological mechanisms and models of FND.

Mechanism/Model	Description	References
**Altered Limbic and Salience Network Activity**	Involves increased limbic/paralimbic activity, heightened amygdala sensitization, and functional connectivity with motor control circuits, suggesting augmented limbic influence over motor behavior.	[61,62,63]
**Self-Agency and Multimodal Integration Disruptions**	Abnormalities in brain activations related to voluntary control perception, contributing to involuntary movement experiences.	[63,64,65,66,67,68,69]
**Attentional and Sensorimotor Circuit Alterations**	Dysregulation in integrating sensory information and emotional processing, particularly involving the insula and cingulate gyrus.	[70,71]
**Cognitive-Behavioral Models**	Suggest symptoms result from subconscious processing of cognitive, emotional, and behavioral factors; heightened anxiety and dissociation play roles in symptom manifestation.	[72,73,74,75]
**Bayesian Model**	Based on predictive coding and active inference; posits that symptoms arise from an imbalance in integrating sensory data and top-down predictions, leading to altered perceptions and motor control.	[76]
**Neurobiologic Models**	Focus on abnormalities in neural networks, particularly involving the orbitofrontal cortex, anterior cingulate cortex, and limbic structures; stress and other factors trigger these networks, leading to symptoms.	[73,77,78,79,80]
**Psychodynamic Models**	Posit that unconscious conflicts manifest as somatic symptoms, serving as a defense mechanism; emphasize the role of early life experiences and abnormal interpersonal relationships.	[81,82,83,84]

**Table 2 ijms-25-04470-t002:** Comparative analysis: functional neurological disorder (FND) vs. Deliberate Symptom Fabrication.

Aspect of Distinction	Functional Neurological Disorder (FND)	Deliberate Symptom Fabrication
**Clinical Presentation**	Characterized by genuine, involuntary symptoms inconsistent with known neurological pathology. Symptoms often fluctuate in severity and distribution over time.	Involves intentional production or exaggeration of symptoms for secondary gain. Symptoms may be consciously simulated or exaggerated, lacking the variability seen in genuine neurological disorders.
**Motivation**	Absence of secondary gain; symptoms are not intentionally produced or maintained for personal benefit.	Involves secondary gain; symptoms are deliberately fabricated or exaggerated for tangible benefits, such as financial compensation, avoiding responsibility, or obtaining attention.
**Response to Suggestion**	Symptoms may show variable responses to suggestion, influenced by psychological factors such as stress or emotional distress.	Symptoms may show less variability or responsiveness to suggestion, as they are consciously controlled by the individual.
**Neuroimaging Findings**	Neuroimaging studies may reveal alterations in brain function or connectivity consistent with functional neurological mechanisms.	Neuroimaging findings may be inconsistent with genuine neurological pathology, suggesting conscious control or a lack of underlying neurological dysfunction.

**Table 3 ijms-25-04470-t003:** Summary of main neuroimaging findings in FND.

Neuroimaging Technique	Description	Contributions to FND Research	Strengths	Limitations
Functional MRI (fMRI)	Measures changes in blood flow to examine brain function.	Investigates functional connectivity patterns and task-based activation in individuals with FND.	Non-invasive, high spatial resolution, can assess dynamic brain activity.	Small sample sizes, variability in imaging protocols and analysis techniques, susceptibility to artifacts, potential influence of confounding factors.
Structural MRI (sMRI)	Assesses brain structure and morphology.	Examines differences in gray matter volume, white matter integrity, and cortical thickness in FND patients compared to healthy controls.	Non-invasive, can provide detailed anatomical information about the brain.	Limited ability to assess dynamic brain changes, potential influence of confounding factors, variability in imaging protocols and analysis techniques.
Diffusion Tensor Imaging (DTI)	Measures diffusion of water molecules to assess white matter microstructure.	Reveals alterations in white matter tracts, indicating disruptions in neural connectivity in FND.	Provides information about white matter integrity and connectivity.	Susceptibility to artifacts, potential influence of confounding factors, variability in imaging protocols and analysis techniques.
Positron Emission Tomography (PET)	Measures brain metabolism and neurotransmitter activity.	Investigates regional cerebral blood flow and glucose metabolism abnormalities in individuals with FND.	Provides complementary information to MRI findings, can assess neurotransmitter function.	Invasive (involves injection of radioactive tracers), lower spatial resolution compared to MRI, limited availability and higher cost, potential influence of confounding factors, radiation exposure.
White Matter Characterization	Assesses white matter structure and integrity.	Identifies alterations in white matter fiber bundles compared to healthy controls.	Provides insights into white matter abnormalities and their contribution to FND pathology.	Susceptibility to artifacts, potential influence of confounding factors, variability in imaging protocols and analysis techniques.
Predictive Neuroimaging	Uses baseline neural activity to predict treatment responses.	Preliminary data suggest predictive value for treatment outcomes, particularly for cognitive behavioral therapy.	Can inform personalized treatment strategies.	Requires further validation and refinement, potential influence of confounding factors, limited availability of predictive models.
Task-Based Neuroimaging	Examines brain activation patterns during specific tasks.	Highlights disturbances in self-agency and motor control networks in FND.	Provides insights into task-specific neural mechanisms underlying FND symptoms.	Susceptibility to task design biases, potential influence of confounding factors, variability in imaging protocols and analysis techniques.
Resting-State Functional Connectivity MRI	Investigates intrinsic brain networks at rest.	Reveals alterations in functional connectivity related to emotion processing and motor control in FND.	Non-invasive, provides information about resting-state brain function.	Potential influence of confounding factors, variability in imaging protocols and analysis techniques.

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
