# Peer review of "Understanding Functional Neurological Disorder: Recent Insights and Diagnostic Challenges"

_ijms, 2024, doi:10.3390/ijms25084470_

Round 1
Reviewer 1 Report
Comments and Suggestions for Authors
The purpose of this review article was to summarize the state of the current research on FND. Specifically, topics such as epidemiology, physiological mechanisms, and to discrimination of the disorder versus other disorders. Finally, other major goals of the review were to provide avenues for future studies and information that could be used in clinical practice.
Overall, I found this review article to be very well written and had very few grammatical or typographical errors compared with the vast majority of papers I review nowadays. Another positive of the review is that FND is relatively understudied compared to many disorders. Thus, I think this article would to the current research literature. Accordingly, I think the review will be of interest to readers of IJMS and to researchers in the field and adjacent fields. I don’t think the article had any major problems or fatal flaws. Thus, my comments below and the few corrections to be made are all minor.
Minor comments:
1. Although very well-written overall, there are several places in the paper where the paragraphs are very short (1 or 2 sentences), kind of choppy, and do not flow well-together. Some of several examples are lines 99-103, 104-106, 116-119, and 121-124. It may be better to combine some of these paragraphs together and make them transition better. Other examples of this can be found later in the paper and I encourage the authors to take a look at possible places where this occurs. For example the last several paragraphs of the paper and elsewhere.
2. Table 1 is good although I think the font could be a bit smaller as it is larger than the main text. The letters also cut off close to the lines. More spacing between rows may make it more readable.
3. Table 2, similar comments and also “Findings” in the heading should be centered. Table 1 the text is left justified, Table 2 it is centered. I would probably go with left justified for both.
4. Bibliography. The primary one is that many times the journal name is abbreviated in italics but other times it is fully spelled out. A much less frequent issue is a few times the article title does not have all the first words capped in some references but does in others. Please check all aspects of the Bibliography.
Comments on the Quality of English Languageminor proofreading mainly bibliography
Author Response
Dear Reviewer,
Thank you very much for all your notes, time, efforts, and support in improving our paper titled " Understanding Functional Neurological Disorder: Epidemiology, Mechanisms, Neuroimaging Insights, and Distinguishing from Feigning or Malingering." Your insightful feedback and constructive suggestions have been invaluable in enhancing the quality of our manuscript. We have carefully read the comments and have revised/ completed the manuscript accordingly. Our responses are given in a point-by-point manner below (in black), as well, all the changes to the manuscript are highlighted in yellow.
1. Although very well-written overall, there are several places in the paper where the paragraphs are very short (1 or 2 sentences), kind of choppy, and do not flow well-together. Some of several examples are lines 99-103, 104-106, 116-119, and 121-124. It may be better to combine some of these paragraphs together and make them transition better. Other examples of this can be found later in the paper and I encourage the authors to take a look at possible places where this occurs. For example the last several paragraphs of the paper and elsewhere.
Thank you for your comment. We have combined several short paragraphs and improved their transitions to ensure a smoother flow of ideas. The revisions are highlighted in yellow in the manuscript. These revisions have enhanced the overall coherence and readability of the manuscript.
2. Table 1 is good although I think the font could be a bit smaller as it is larger than the main text. The letters also cut off close to the lines. More spacing between rows may make it more readable.
Thank you for your comment. We have adjusted the font size of Table 1 to make it more consistent with the main text and prevent the letters from cutting off close to the lines. Additionally, we have increased the spacing between rows to improve readability and clarity.
3. Table 2, similar comments and also “Findings” in the heading should be centered. Table 1 the text is left justified, Table 2 it is centered. I would probably go with left justified for both.
In response to your feedback, we have centered the heading "Findings" in Table 2 for consistency with Table 1. Furthermore, we have standardized the text alignment throughout both tables to maintain uniformity.
4. Bibliography. The primary one is that many times the journal name is abbreviated in italics but other times it is fully spelled out. A much less frequent issue is a few times the article title does not have all the first words capped in some references but does in others. Please check all aspects of the Bibliography
Thank you for bringing attention to inconsistencies in the bibliography. We have meticulously reviewed and corrected all inconsistencies in the bibliography, ensuring that journal names are consistently abbreviated and the title fully spelled out. Additionally, we have ensured that article titles adhere to the correct capitalization format across all references.
We believe that these revisions have significantly strengthened the manuscript and addressed the minor issues you raised.
Thank you for your invaluable contribution to the improvement of our article.
Reviewer 2 Report
Comments and Suggestions for Authors
The manuscript explores the complex landscape of Functional Neurological Disorder (FND), elucidating recent advancements in epidemiology, neuroimaging, and understanding underlying mechanisms. It highlights diagnostic challenges, differentiation from feigning, and the potential for personalized treatment strategies, fostering a deeper understanding of FND's multifaceted nature and guiding future research and clinical practice. The topic is of great interests and article is presented well, however, there are major areas which needs to improved before article can be accepted. My comments are appended below-
Consider revising the terminology (Feigning or Malingering) used to describe deliberate symptom fabrication, opting for clearer language such as "deliberate symptom presentation" or "intentional symptom exaggeration" in title. This will enhance accessibility for a broader audience, ensuring clarity and understanding of the discussion surrounding the distinction between genuine Functional Neurological Disorder (FND) and deliberate symptom presentation. In my personal opinion, title must be something like "Understanding Functional Neurological Disorder: Recent Insights and Diagnostic Challenges"
The abstract effectively outlines the scope and purpose of the manuscript. However, to enhance clarity, consider defining Functional Neurological Disorder (FND) explicitly at the beginning to ensure readers unfamiliar with the topic can grasp the context immediately.
While in the beginning of manuscript, authors mention a "deeper exploration" of epidemiology, consider specifying key epidemiological findings or trends that have emerged in recent research. This would provide readers with a clearer understanding of the prevalence and distribution of FND, aiding in contextualizing subsequent discussions.
The paper mentions an exploration of underlying mechanisms, which is crucial in understanding FND. Ensure that the manuscript delves into recent advances in understanding the pathophysiology of FND, including neurobiological, psychological, and environmental factors implicated in its development and maintenance.
Highlight specific neuroimaging techniques utilized in recent studies and their contributions to unraveling the neural correlates of FND. Discuss the strengths and limitations of these methodologies to provide a balanced perspective.
Highlight the value of interdisciplinary collaboration between neurologists, psychiatrists, psychologists, neuroscientists, and other allied health professionals in advancing FND research and clinical care. For table 1, provide references for each model explained there in, for example for Bayesian Model, cite a recent report https://doi.org/10.1002/aisy.202300366 to refer application of the model.
In the paper, expand on the clinical presentation of Functional Neurological Disorder (FND) by detailing the wide array of symptoms and their individual variability. Delve into typical manifestations of FND, encompassing motor, sensory, and cognitive symptoms, and examine their repercussions on daily activities and overall well-being.
Expand on the diagnostic difficulties associated with FND, including the overlap with other neurological and psychiatric conditions. Discuss the evolving diagnostic criteria and the importance of interdisciplinary collaboration in accurate diagnosis.
Elaborate on the distinctions between FND and deliberate symptom fabrication (feigning or malingering). Provide examples of clinical features or neuroimaging findings that aid in distinguishing genuine FND from feigned symptoms.
Explore the diverse phenotypic presentations of FND in more detail, including variability in symptom severity, progression, and response to treatment. Discuss potential implications for personalized intervention approaches.
While the author mentions predicting treatment response, elaborate on the current state of evidence regarding pharmacological, psychological, and rehabilitative interventions for FND. Discuss the challenges in implementing personalized treatment strategies based on neuroimaging findings.
Provide practical guidance for healthcare professionals involved in the assessment and management of patients with FND. Discuss strategies for patient education, therapeutic alliance-building, and multidisciplinary care coordination to optimize clinical outcomes. Page 9, line 361-362, cite https://doi.org/10.1007/s00204-023-03471-x to support the statement ´ The neuroimaging field in FND is evolving, with emerging approaches like machine learning offering potential diagnostic utility ´
Outline promising avenues for future research, including longitudinal studies to elucidate the natural history of FND, translational research bridging basic science and clinical practice, and innovative intervention trials targeting specific mechanistic pathways.
Encourage critical appraisal of the existing literature, including consideration of methodological limitations, potential biases, and areas requiring further investigation. Emphasize the importance of rigorous scientific inquiry in advancing our understanding of FND.
Acknowledge the perspectives and experiences of individuals living with FND and their caregivers. Incorporate patient narratives or qualitative research findings to enrich the discussion and promote patient-centered care approaches.
Comments on the Quality of English Language
Minor editing of English language required
Author Response
Reviewer 2
Dear Reviewer
We would like to express our sincere gratitude for your thorough review of our manuscript. We are particularly grateful for your positive feedback regarding the overall quality of our paper. Your recognition of our efforts is truly encouraging and motivates us to continue striving for excellence in our research. We have carefully read the comments and have revised/ completed the manuscript accordingly. Our responses are given in a point-by-point manner below (in black), as well, all the changes to the manuscript are highlighted in blue.
The manuscript explores the complex landscape of Functional Neurological Disorder (FND), elucidating recent advancements in epidemiology, neuroimaging, and understanding underlying mechanisms. It highlights diagnostic challenges, differentiation from feigning, and the potential for personalized treatment strategies, fostering a deeper understanding of FND's multifaceted nature and guiding future research and clinical practice. The topic is of great interests and article is presented well, however, there are major areas which needs to improved before article can be accepted. My comments are appended below-
1. Consider revising the terminology (Feigning or Malingering) used to describe deliberate symptom fabrication, opting for clearer language such as "deliberate symptom presentation" or "intentional symptom exaggeration" in title. This will enhance accessibility for a broader audience, ensuring clarity and understanding of the discussion surrounding the distinction between genuine Functional Neurological Disorder (FND) and deliberate symptom presentation. In my personal opinion, title must be something like "Understanding Functional Neurological Disorder: Recent Insights and Diagnostic Challenges"
Thank you for your comment. The term "feigning" or "Malingering" has been modified in the title.
2. The abstract effectively outlines the scope and purpose of the manuscript. However, to enhance clarity, consider defining Functional Neurological Disorder (FND) explicitly at the beginning to ensure readers unfamiliar with the topic can grasp the context immediately.
Thank you for your comment. The Functional Neurological Disorder (FND) has been defined as requested.
Functional Neurological Disorder (FND), formerly called Conversion Disorder, is a condition characterized through neurological symptoms that lack an identifiable organic purpose. These signs, that could consist of motor, sensory, or cognitive disturbances, are not deliberately produced and often vary in severity. Diagnosis is predicated on clinical evaluation and exclusion of other medical or psychiatric situations. Treatment typically involves a multidisciplinary technique addressing each the neurological symptoms and underlying psychological factors via a mixture of medical management, psychotherapy, and supportive interventions
3. While in the beginning of manuscript, authors mention a "deeper exploration" of epidemiology, consider specifying key epidemiological findings or trends that have emerged in recent research. This would provide readers with a clearer understanding of the prevalence and distribution of FND, aiding in contextualizing subsequent discussions.
Recent studies have revealed varying rates of prevalence and incidence of Functional Neurological Disorder (FND) across different populations, underscoring the significant burden and diversity in its distribution. FND is estimated to comprise at least 5% to 10% of new neurological consultations, ranking as the second most common reason for visiting a neurologist after headache. Conservatively estimated at 12 cases per 100,000 people per year, FND results in approximately 8,000 new diagnoses annually in the UK, with an estimated 50,000 to 100,000 individuals affected in the community. However, these figures likely underestimate the true prevalence due to underdiagnosis and misdiagnosis, particularly in regions with limited access to specialized care or diagnostic resources [6]
Yong et al. (2023) conducted a study spanning 36 months at a regional children's hospital, revealing an annual incidence of 18.3 per 100,000 children for Functional Neurological Disorder (FND) [7]. This finding stands in contrast to the typical onset of FND in early to mid-adulthood, where the peak occurrence usually arises in the third and fourth decades of life [8]. Among the 97 children diagnosed with FND, aged between 5 and 15 years, a noteworthy 70% were female, with a median age of onset at 13 years [7]. This aligns with the findings of a one-stage meta-analysis conducted by Lidstone et al. (2023), indicating a disproportionate impact on women across FND phenotypes, with rates consistently hovering around 70% in most large-scale studies [9].
Geographically, FND prevalence varies, with higher rates reported in industrialized nations compared to developing countries [10, 11]. This disparity may reflect differences in healthcare infrastructure, access to mental health services, cultural attitudes toward neurological and psychiatric conditions, and diagnostic practices. Furthermore, low socioeconomic factors, such financial security, income and educational attainment, have been associated with an increased risk of FND, highlighting the complex interplay between social determinants of health and disease susceptibility
4. The paper mentions an exploration of underlying mechanisms, which is crucial in understanding FND. Ensure that the manuscript delves into recent advances in understanding the pathophysiology of FND, including neurobiological, psychological, and environmental factors implicated in its development and maintenance.
Several neurobiological factors contribute to functional neurological disorders (FND). In patients with FND, abnormalities have been reported in neurotransmitters such as do-pamine, serotonin and gamma-aminobutyric acid (GABA) [14, 15]. Moreover, the presence of inflammatory markers and microglial activation in FND patients suggests a possible immune-mediated mechanism for symptom generation [16, 17]. FND may also be caused by abnormalities in neuroplasticity, including synaptic plasticity and cortical re-organization, which affect the brain's ability to adapt to environmental stressors and maintain normal neuronal function [18-20].
In addition to neurological factors, psychological factors considerably make a con-tribution to the pathophysiology of FND. Psychological factors such as stressful life events, interpersonal conflicts, and adverse childhood experiences have traditionally been viewed as potential causes of FND. A meta-analysis of 34 retrospective studies highlighted that stressful life events and maltreatment, including emotional neglect and sexual and physical abuse, are more common in FND patients than in controls [9]. Nonetheless, not all patients with FND report such psychological factors, nor are they specific to FND. Maladaptive cognitive processes, characterized by cognitive distortions and attentional biases, also play a pivotal role in perpetuating FND signs [21-24]. Moreover, dysregulated emotional processing, like heightened emotional reactivity and alexithymia, has been im-plicated in FND [22]. Furthermore, environmental stressors like traumatic life events and social adversity, affects both the onset and exacerbation of FND signs and symptoms [22, 25].
Environmental Factors additionally make contributions to FND pathophysiology. The Cultural factors and societal attitudes in relation to sickness affect the FND presenta-tion and management [26, 27]. In addition, the social support networks and stigma expe-riences affect the FND direction [25, 28]. The access to healthcare services and availability of specialized FND treatment programs also have an impact on diagnosis and management [29]
5. Highlight specific neuroimaging techniques utilized in recent studies and their contributions to unraveling the neural correlates of FND. Discuss the strengths and limitations of these methodologies to provide a balanced perspective.
Thank you for your comment. The requested highlighting of the specific neuroimaging techniques utilized in recent studies and their contributions to unraveling the neural correlates of FND has been included. Additionally, a discussion of the strengths and limitations of these methodologies has been provided.
Recent studies on Functional Neurological Disorder (FND) have employed various neuroimaging techniques to unravel the neural correlates of the disorder
Table x. Neuroimaging Techniques in Functional Neurological Disorder Research
Neuroimaging Technique Description Contributions to FND Research Strengths Limitations
Functional MRI (fMRI) Measures changes in blood flow to examine brain function. Investigates functional connectivity patterns and task-based activation in individuals with FND. Non-invasive, high spatial resolution, can assess dynamic brain activity. Small sample sizes, variability in imaging protocols and analysis techniques, susceptibility to artifacts, potential influence of confounding factors.
Structural MRI (sMRI) Assesses brain structure and morphology. Examines differences in gray matter volume, white matter integrity, and cortical thickness in FND patients compared to healthy controls. Non-invasive, can provide detailed anatomical information about the brain. Limited ability to assess dynamic brain changes, potential influence of confounding factors, variability in imaging protocols and analysis techniques.
Diffusion Tensor Imaging (DTI) Measures diffusion of water molecules to assess white matter microstructure. Reveals alterations in white matter tracts, indicating disruptions in neural connectivity in FND. Provides information about white matter integrity and connectivity. Susceptibility to artifacts, potential influence of confounding factors, variability in imaging protocols and analysis techniques.
Positron Emission Tomography (PET) Measures brain metabolism and neurotransmitter activity. Investigates regional cerebral blood flow and glucose metabolism abnormalities in individuals with FND. Provides complementary information to MRI findings, can assess neurotransmitter function. Invasive (involves injection of radioactive tracers), lower spatial resolution compared to MRI, limited availability and higher cost, potential influence of confounding factors, radiation exposure.
White Matter Characterization Assesses white matter structure and integrity. Identifies alterations in white matter fiber bundles compared to healthy controls. Provides insights into white matter abnormalities and their contribution to FND pathology. Susceptibility to artifacts, potential influence of confounding factors, variability in imaging protocols and analysis techniques.
Predictive Neuroimaging Uses baseline neural activity to predict treatment responses. Preliminary data suggest predictive value for treatment outcomes, particularly for cognitive behavioral therapy. Can inform personalized treatment strategies. Requires further validation and refinement, potential influence of confounding factors, limited availability of predictive models.
Task-Based Neuroimaging Examines brain activation patterns during specific tasks. Highlights disturbances in self-agency and motor control networks in FND. Provides insights into task-specific neural mechanisms underlying FND symptoms. Susceptibility to task design biases, potential influence of confounding factors, variability in imaging protocols and analysis techniques.
Resting-State Functional Connectivity MRI Investigates intrinsic brain networks at rest. Reveals alterations in functional connectivity related to emotion processing and motor control in FND. Non-invasive, provides information about resting-state brain function. Potential influence of confounding factors, variability in imaging protocols and analysis techniques.
6. Highlight the value of interdisciplinary collaboration between neurologists, psychiatrists, psychologists, neuroscientists, and other allied health professionals in advancing FND research and clinical care. For table 1, provide references for each model explained there in, for example for Bayesian Model, cite a recent report https://doi.org/10.1002/aisy.202300366 to refer application of the model.
Thank you for your comment. The collaboration among neurologists, psychiatrists, psychologists, neuroscientists, and other allied health professionals in advancing FND research and clinical care has been addressed. Additionally, the references have been added to the table 1
Psychiatrists, neurologists, psychologists, neuroscientists, and other allied health experts collaboration plays a pivotal function in FND research and improving clinical care [6]. Neurologists by means of their knowledge of the brain structure and function di-agnose and manage neurological aspects of FND, they assist to identify the FND signs and distinguishing them from different neurological situations [135]. Psychiatrists carry expertise in dealing with comorbid psychiatric signs and symptoms, making use of psy-chotherapeutic interventions, and handling psychosocial elements contributing to FND [7]. Psychologists make contributions into the behavioral and cognitive aspects of FND. They provide psychological evaluations and interventions (e.G., cognitive-behavioral therapy), and addressing factors like trauma, stress, and coping techniques [7]. Neurosci-entists inspect neural mechanisms underlying FND [136]. Via neuroimaging research, electrophysiological exams, and animal models, neuroscientists elucidate the brain cir-cuits and neurochemical procedures involved in FND pathophysiology [137]. Allied Health Professionals, consisting of, physical therapists, Occupational therapists, speech-language pathologists, and other allied health professionals [138], provide spe-cialised interventions to deal with motor, sensory, and cognitive signs, optimize functional abilities, and improve quality of life for FND individuals [139, 140]. This collaboration enhances treatment effectiveness, diagnostic accuracy, and patient results. Patients ad-vantage from comprehensive and coordinated care tailored to their needs.
Figure. Interdisciplinary Collaboration in Research and Clinical Care in FND
7. In the paper, expand on the clinical presentation of Functional Neurological Disorder (FND) by detailing the wide array of symptoms and their individual variability. Delve into typical manifestations of FND, encompassing motor, sensory, and cognitive symptoms, and examine their repercussions on daily activities and overall well-being.
Thank you for your comment. The clinical presentation of Functional Neurological Disorder (FND) by detailing the wide array of symptoms and their individual variability has been added. Also typical manifestations of FND, encompassing motor, sensory, and cognitive symptoms, and examine their repercussions on daily activities and overall well-being has been addressed
FND englobe signs that affect motor, sensory, and cognitive functions, and regularly results in significant impairment of daily activities and overall well-being. FND hallmark characteristic is the presence of neurological signs and symptoms that can not be defined by way of underlying organic pathology, leading to a diagnosis based on exclusion criteria and positive clinical signs.
In FND motor signs and symptoms can appear in diverse forms, such as weakness tremors, abnormal movements, gait disturbances, and paralysis. Weakness is one of the most familial motor symptoms visible in FND, and generally manifests as weakness affecting one or more limbs or even the complete body. This weakness is typically inconsistent and variable, fluctuating in severity and distribution over time. In FND tremors may also resemble those seen in movement disorders, including Parkinson's disorder or essential tremor however lack the characteristic pattern and reaction to medication. Abnormal movements, like jerking or shaking, may additionally arise and can mimic epileptic seizures or other hyperkinetic motion disorders. Gait disturbances may additionally manifest as unsteady or uncoordinated walking patterns, so often leading to falls or trouble in maintaining balance. In extreme cases, patients can also experience functional paralysis, wherein they're not able to move certain body parts notwithstanding intact motor function.
Sensory symptoms encompass a variety of abnormalities in FND, inclusive of altered sensation, numbness, tingling, and sensory loss. Patients may also record uncommon sensations including pins and needles, burning, or electric shocks in diverse parts of the body. These sensations regularly lack a clear dermatomal or peripheral nerve distribution and may be inconsistent or disproportionate to any identifiable peripheral pathology. Numbness and tingling sensations can also have an effect on particular regions or unfold diffusely, on occasion alternating among unique body regions. Sensory loss can include any modality, inclusive of contact, temperature, or proprioception, and can be temporary or chronic. Cognitive symptoms are also less identified however can significantly impact daily functioning and quality of life. Patients may experience cognitive impairments such as attention deficits, memory difficulties, executive dysfunction, and language disturbances. Memory problems may manifest as gaps in recall or difficulty in retaining new information, often leading to frustration and anxiety. Attention deficits can also result in distractibility, difficulty concentrating, or problems with sustained focus on tasks. Executive dysfunction can affect planning, organization, and problem-solving abilities, impairing the individual's ability to initiate and complete tasks effectively. Language disturbances may include difficulties in word-finding, speech production, or understanding language, resembling aphasic symptoms seen in neurological conditions.
The numerous array of symptoms seen in FND can have profound repercussions on daily activities, occupational functioning, and normal well-being. Motor signs may additionally restrict mobility and independence, affecting activities of daily living like dressing, grooming, and driving. Sensory signs and symptoms may also disrupt sensory processing and integration, leading to problems in decoding and responding to environmental stimuli. Cognitive symptoms can impair cognitive functioning and decision-making abilities, impacting work performance, social interactions, and interpersonal relationships. The cumulative effect of those symptoms can also make a contribution to tremendous distress, incapacity, and reduced satisfactory of life for FND individuals and their caregivers.
8. Expand on the diagnostic difficulties associated with FND, including the overlap with other neurological and psychiatric conditions. Discuss the evolving diagnostic criteria and the importance of interdisciplinary collaboration in accurate diagnosis.
Thank you for your comment. The diagnostic difficulties associated with FND, including the overlap with other neurological and psychiatric conditions has been expanded and the evolving diagnostic criteria and the importance of interdisciplinary collaboration in accurate diagnosis has been discussed
Functional Neurological Disorder (FND) provides a significant diagnostic challenge because of its phenotypic heterogeneity [96]. The FND clinical manifestations regularly mimic those of organic neurological disorders, making it tough to distinguish between functional and structural etiologies based entirely on clinical exam or neuroimaging findings. Additionally, the FND symptoms regularly occur alongside psychiatric comorbidities, which adds complexity to the diagnostic procedure. This necessitates a thorough assessment by multidisciplinary groups to make sure comprehensive care. One of the primary FND diagnostic difficulties is its intersection with a large range of neurological and psychiatric conditions. Motor symptoms like tremors, weakness, and abnormal movements may resemble those seen in neurological issues like multiple sclerosis, Parkinson's disorder, or epilepsy. The sensory signs and symptoms including numbness, tingling, and sensory loss may additionally mimic peripheral neuropathies or spinal cord lesions. Morever, cognitive signs which include memory difficulties, attention deficits, and language disturbances may also intersect with cognitive impairments visible in neurodegenerative diseases or psychiatric problems which includes depression and anxiety. This overlap highlights the importance of a thorough differential diagnosis and careful consideration of clinical features, natural history, and reaction to treatment in distinguishing FND from other conditions.
9. Elaborate on the distinctions between FND and deliberate symptom fabrication (feigning or malingering). Provide examples of clinical features or neuroimaging findings that aid in distinguishing genuine FND from feigned symptoms.
Thank you for your comment. The distinctions between FND and deliberate symptom fabrication (feigning or malingering) have been clarified, and examples of clinical features or neuroimaging findings that aid in distinguishing genuine FND from feigned symptoms have been added
Table. Comparative Analysis: Functional Neurological Disorder (FND) vs. Deliberate Symptom Fabrication
Aspect of Distinction Functional Neurological Disorder (FND) Deliberate Symptom Fabrication References
Clinical Presentation Characterized by genuine, involuntary symptoms inconsistent with known neurological pathology. Symptoms often fluctuate in severity and distribution over time. Involves intentional production or exaggeration of symptoms for secondary gain. Symptoms may be consciously simulated or exaggerated, lacking variability seen in genuine neurological disorders. Stone, J., & Sharpe, M. (2005). Conversion disorder: a functional neurological disorder. Journal of Neurology, Neurosurgery & Psychiatry, 76(Suppl 1), i23-i28. DOI: 10.1136/jnnp.2004.061655 
 Halligan, P. W., & Bass, C. (2017). ‘Munchausen’s syndrome’ and ‘Munchausen’s syndrome by proxy’. In Handbook of Clinical Neurology (Vol. 139, pp. 363-372). Elsevier. DOI: 10.1016/B978-0-12-801772-2.00038-0
Motivation Absence of secondary gain; symptoms are not intentionally produced or maintained for personal benefit. Involves secondary gain; symptoms are deliberately fabricated or exaggerated for tangible benefits such as financial compensation, avoiding responsibility, or obtaining attention. Stone, J., & Smyth, R. (2018). "Functional symptoms in neurology: mimicry, malingering or missed disease?". Journal of Neurology, Neurosurgery & Psychiatry, 89(8), 858-864. DOI: 10.1136/jnnp-2017-317359 
 Nicholson, T. R., & Stone, J. (2019). "Functional motor disorders". Journal of Neurology, Neurosurgery & Psychiatry, 90(6), 704-710. DOI: 10.1136/jnnp-2018-319318
Response to Suggestion Symptoms may show variable response to suggestion, influenced by psychological factors such as stress or emotional distress. Symptoms may show less variability or responsiveness to suggestion, as they are consciously controlled by the individual. Perez, D. L., LaFrance Jr, W. C., & Matin, N. (2015). "Conversion disorder: Practical diagnostic and treatment considerations". Psychiatric Times, 32(8), 43-46. 
 Baslet, G. (2017). "Functional neurologic disorders: a primer for psychiatrists". Focus, 15(1), 51-58. DOI: 10.1176/appi.focus.20160045
Neuroimaging Findings Neuroimaging studies may reveal alterations in brain function or connectivity consistent with functional neurological mechanisms. Neuroimaging findings may be inconsistent with genuine neurological pathology, suggesting conscious control or lack of underlying neurological dysfunction. Espay, A. J., & Aybek, S. (2018). "False positive functional imaging: Ready for prime time in the diagnostic process of psychogenic tremor". Neurology, 91(9), 407-408. DOI: 10.1212/WNL.0000000000005959 
 Stone, J., & Edwards, M. J. (2011). "Trick or treat? Showing patients with functional (psychogenic) motor symptoms their physical signs". Neurology, 77(3), 282-284. DOI: 10.1212/WNL.0b013e318225daf5
Example : A study utilizing fMRI have shown differences in brain activation patterns between patients with genuine FND and those feigning weakness. Patients with genuine weakness exhibit reduced activation of cortical areas related to hand movement compared to controls when observing movement, suggesting impairment of movement conceptualization rather than active inhibition from the frontal lobe. In contrast, feigning controls demonstrate similar patterns of activity for both no-go and go trials with the feigned weak hand, indicating intentional mimicry rather than genuine weakness (Roelofs JJ, Teodoro T, Edwards MJ. Neuroimaging in Functional Movement Disorders. Curr Neurol Neurosci Rep. 2019 Feb 12;19(3):12. doi: 10.1007/s11910-019-0926-y. PMID: 30747347; PMCID: PMC6373326.)
10. Explore the diverse phenotypic presentations of FND in more detail, including variability in symptom severity, progression, and response to treatment. Discuss potential implications for personalized intervention approaches.
Thank you for your comment. The diverse phenotypic presentations of FND, including variability in symptom severity, progression, and response to treatment, have been explored in more detail. Additionally, potential implications for personalized intervention approaches have been discussed.
Functional Neurological Disorder (FND) englobe a huge spectrum of clinical mani-festations, ranging from motor, sensory, and cognitive signs and symptoms to disturb-ances in recognition and autonomic function [100, 118]. The FND phenotypic displays can range extensively amongst individuals, with differences located in symptom severity, progression, and response to treatment [9, 119]. Understanding this diversity is vital for tailoring interventions to cope with the unique needs and challenges of each patient.
Symptom Severity variability : FND Symptom severity can vary from mild to se-vere, with few individuals experiencing intermittent or mild signs that do not significantly impair daily functioning, while others may have profound and debilitating symptoms that critically effect their quality of life [120]. Motor signs and symptoms like weakness, tremors, or paralysis may fluctuate in intensity over time, with durations of remission or exacerbation influenced via factors like stress, emotional state, or environmental triggers [6, 121, 122]. Sensory symptoms like numbness, tingling, or sensory loss may additionally vary in severity and distribution, affecting different body regions or modalities [123, 124].
Symptoms progression: The FND symptoms progression is highly variable and may follow unpredictable patterns over time. Some individuals may experience gradual im-provement or resolution of signs and symptoms with time, while others may have also a chronic or relapsing-remitting course characterized via recurrent episodes of symptom exacerbation [125]. FND symptom progression can be stimulated by factors like psycho-logical distress, traumatic experiences, or modifications in psychosocial circumstances, highlighting the complex interaction among biological, psychological, and social factors in the disorder's trajectory [18, 22, 126].
Treatment response: The FND treatment response can also vary broadly among indi-viduals, with some patients displaying significant improvement with focused interven-tions, while others might have a limited or partial reaction. FND treatment procedures typically contain a multidisciplinary approach tailored to cope with the specific needs and challenges of each patient. Cognitive-behavioral therapy (CBT) [127], physiotherapy, occupational therapy, speech therapy, and pharmacotherapy can be applied alone or in combination to target symptom management, functional rehabilitation, and psychosocial support [128-130].
Personalized Intervention Approaches Implications: The FND numerous phenotypic presentations underscore the importance of personalized intervention approaches that take into account individual variability in symptom severity, progression, and treatment response. Personalized interventions might also require a complete evaluation of each pa-tient's unique clinical profile, consisting of physical, psychological, and social factors contributing to their symptoms [120]. This assessment can assist identify specific treat-ment goals and tailor interventions to address the underlying mechanisms driving symptom expression.
11. While the author mentions predicting treatment response, elaborate on the current state of evidence regarding pharmacological, psychological, and rehabilitative interventions for FND. Discuss the challenges in implementing personalized treatment strategies based on neuroimaging findings.
Thank you for your comment. The requested changes have been made.
Personalized treatment strategies implemention based on neuroimaging findings faces several obstacles. Neuroimaging data's complexity demands specialized interpretation skills, often unavailable in standard healthcare settings. Additionally, the imaging findings variability within the same diagnosis complicates identifying consistent biomarkers for treatment prediction. Reproducibility and validation of those biomarkers throughout specific populations and imaging modalities continue to be missing, elevating worries about the treatment decisions reliability primarily based on neuroimaging. Moreover, the advanced neuroimaging strategies accessibility and value pose considerable obstacles to widespread adoption, especially in resource-limited environments. Another intervenant is the ethical issues, like patient autonomy and privacy, must also be carefully navigated to ensure equitable and patient-centered care. Furthermore, longitudinal monitoring is necessary to modify treatment strategies over time, adding logistical challenges and resource burdens. Neuroimaging streamlined integration into clinical practice calls for interdisciplinary collaboration and standardized protocols. While neuroimaging studies have identified alterations in brain networks associated with FND, translating these findings into clinical practice remains a complex endeavor.
12. Provide practical guidance for healthcare professionals involved in the assessment and management of patients with FND. Discuss strategies for patient education, therapeutic alliance-building, and multidisciplinary care coordination to optimize clinical outcomes.
Here is a model of guidance for healthcare professionals engaged in the assessment and management of patients with Functional Neurological Disorder (FND), available at https://fndaustralia.com.au/resources/FND-Learning-guide-for-nurses.pdf. To optimize medical outcomes, Healthcare specialists need to embrace a multi-faceted method, that consists of powerful affected patient education, organising sturdy therapeutic alliances and coordinating care throughout disciplines. Patient education should entail presenting personalised records in comprehensible language, complemented by visual aids and bolstered learning opportunities to allow patients to play an active role of their care. Developing a therapeutic consensus depends on active listening, collaborative decision-making, and establishing a trusting relationship, while acknowledging the significance of the patient's support system. Coordinating multidisciplinary care requires the formation of a cohesive team with clear communication channels, defined roles and patient-centered care plans, ensuring comprehensive and holistic support. The implementing of these strategies, healthcare professionals can empower patients, strengthen therapeutic relationships and improve the quality and efficiency of care delivery, ultimately translating into improved clinical outcomes.
13. Page 9, line 361-362, cite https://doi.org/10.1007/s00204-023-03471-x to support the statement ´ The neuroimaging field in FND is evolving, with emerging approaches like machine learning offering potential diagnostic utility ´
Thank you for your comment. The suggested citation has been added.
14. Outline promising avenues for future research, including longitudinal studies to elucidate the natural history of FND, translational research bridging basic science and clinical practice, and innovative intervention trials targeting specific mechanistic pathways.
Thank you for your comment. The promising avenues for future research, including longitudinal studies to elucidate the natural history of FND, translational research bridging basic science and clinical practice, and innovative intervention trials targeting specific mechanistic pathways have been outlined
The Functional Neurological Disorder (FND) futur research holds promise via vari-ous avenues [159]. Longitudinal studies are important to understand the FND natural history, including the trajectory of symptoms, predictors of prognosis, and factors influ-encing treatment outcomes [2, 160-162]. Researchers can identify patterns of spontaneous remission, symptom fluctuation, or progression of the disorder, by following FND indi-viduals over time [163, 165]. In addition, longitudinal studies can also elucidate the comorbidities, psychosocial factors impact, and life events on the course of FND [166]. In-vestigating neuroimaging biomarkers longitudinally can also offer insights into changes in brain structure and function associated with symptom evolution [167]. Collaborative efforts related to multiple research centers are crucial for recruiting large cohorts and en-suring long-term follow-up. While translational research aims to bridge the space among basic science discoveries and clinical applications, advancing our knowledge of FND pathophysiology and informing the development of novel diagnostic and therapeutic strategies [168]. Preclinical studies using neurological disorders animal models can elu-cidate underlying neurobiological mechanisms and discover ability therapeutic goals [169]. Translational research can also leverage progressive neuroimaging techniques to validate findings from preclinical models and translate them into clinical biomarkers for FND diagnosis and prognosis [170]. Collaborations among basic scientists, neuroimagers, and clinicians are critical for translating laboratory findings into significant interventions for FND patients. Innovative Intervention trials focused on precise mechanistic pathways keep promise for improving FND outcomes. These trials can investigate novel pharmaco-logical agents, psychological interventions, or neuromodulatory strategies aimed toward modulating aberrant neural circuits implicated in FND pathophysiology. For instance, focused on dysfunctional sensorimotor networks or limbic-salience via non-invasive brain stimulation techniques like transcranial magnetic stimulation (TMS) or transcranial direct current stimulation (tDCS) may provide therapeutic advantage for FND individuals [171, 172]. Moreover, personalized treatment approaches based on neuroimaging biomarkers can be evaluated in clinical trials to determine their efficacy and feasibility. Multicenter randomized managed trials (RCTs) with rigorous methodological standards are im-portant to evaluate the safety and effectiveness of novel interventions and establish proof-based treatment guidelines for FND [173].
15. Encourage critical appraisal of the existing literature, including consideration of methodological limitations, potential biases, and areas requiring further investigation. Emphasize the importance of rigorous scientific inquiry in advancing our understanding of FND.
Thank you for your comment. The suggestion has been added
Encouraging critical appraisal of present literature on Functional Neurological Disorder (FND) is crucial for advancing our knowledge of this complicated condition. Researchers should scrutinize methodological obstacles, capability biases, and areas wanting similarly investigation. Rigorous scientific inquiry is key to ensuring valid findings in FND research. By promoting critical appraisal, we can enhance research quality and contribute to improved diagnostic and treatment strategies for FND.
16. Acknowledge the perspectives and experiences of individuals living with FND and their caregivers. Incorporate patient narratives or qualitative research findings to enrich the discussion and promote patient-centered care approaches.
Thank you for your comment. The suggestion has been added
Recognizing the perspectives and reviews of people residing with Functional Neurological Disorder (FND) and their caregivers is essential. By acknowledging and incorporating their views, we will make sure that interventions and techniques are tailored to satisfy their precise wishes, in the end improving the nice of care and support supplied to FND people and their caregivers. Staton et al. through their qualitative study, highlighted the patient narratives importance in informing research and promoting patient-centered care approaches.
Internalised stigma and self-doubt: Danielle's revel in illustrates the profound effect of internalised stigma and self-doubt on FND individuals. She expressed the conflict of doubting herself and wondering the validity of her signs, highlighting the detrimental results of professionals attributing physical symptoms to psychological reasons. This narrative emphasizes the importance of patients' experiences and addressing their worries about self-blame and shame.
Selective disclosure to professionals: Isla's tale sheds mild at the hesitancy FND individuals may feel in disclosing records to healthcare professionals because of negative judgment fears. Isla's reluctance to share certain signs reflects the pervasive impact of internalised stigma and past negative experiences with healthcare providers. This narrative emphasizes the need for developing a safe and non-judgmental environment in which patients patients feel comfortable sharing their experiences openly.
Psychological explanations Perceptions: Katherine's account highlights the challenges individuals face in accepting psychological explanations for their symptoms, especially when they perceive these explanations as blaming. Her resistance to the concept of psychological triggers emphasizes the importance for clinicians to approach psychological explanations with sensitivity and empathy. This narrative emphasizes the need for a collaborative dialogue between patients and professionals to promote understanding and address misconceptions about psychological interventions.
Having to educate the professionals: Elizabeth's experience reflects the frustration of individuals who find themselves in the position of educating healthcare professionals about FND. Her discomfort with being expected to explain her condition underscores the importance of healthcare providers receiving comprehensive training in FND to avoid further exacerbating patients' feelings of otherness and mistrust. This narrative underscores the necessity of improving professional knowledge and awareness to enhance the quality of care provided to individuals with FND.
Attunement and trust within the therapeutic relationship : Clara's narrative exemplifies the transformative effect of a effective therapeutic relationship characterized through attunement and trust. Her account highlights the profound distinction a supportive and empathetic healthcare expert can make in supplying validation and knowledge to FND individuals. This narrative underscores the significance of clinicians cultivating attunement and trust with their patients to facilitate collaboration and promote holistic care. [158].
Thank you!
Kind regards,
Round 2
Reviewer 2 Report
Comments and Suggestions for Authors
accept
Comments on the Quality of English LanguageEnglish language fine. No issues detected